# Research

ecology, environmental science, plant science

photosystem disruption, microarthropod abundance, apex predator, ecosystem heating, mesocosm, body size

**Author for correspondence:**
Adam J. Vanbergen
e-mail: adam.vanbergen@inrae.fr

# Habitat loss, predation pressure and episodic heat-shocks interact to impact arthropods and photosynthetic functioning of microecosystems

Adam J. Vanbergen[1,3], Claire Boissieres[2], Alan Gray[3] and Daniel S. Chapman[3,4]

[1]Agroécologie, AgroSup Dijon, INRAE, Univ. Bourgogne Franche-Comté, F-21000 Dijon, France
[2]L'Ecole Nationale Supérieure Agronomique de Toulouse (ENSAT), Avenue de l'Agrobiopole, BP 32607, Auzeville-Tolosane 31326, Castanet-Tolosan, France
[3]UK Centre for Ecology and Hydrology, Bush Estate, Penicuik, Midlothian EH26 0QB, UK
[4]Biological and Environmental Sciences, University of Stirling, Stirling FK9 4LA, UK

AJV, 0000-0001-8320-5535

Ecosystems face multiple, potentially interacting, anthropogenic pressures that can modify biodiversity and ecosystem functioning. Using a bryophyte–microarthropod microecosystem we tested the combined effects of habitat loss, episodic heat-shocks and an introduced non-native apex predator on ecosystem function (chlorophyll fluorescence as an indicator of photosystem II function) and microarthropod communities (abundance and body size). The photosynthetic function was degraded by the sequence of heat-shock episodes, but unaffected by microecosystem patch size or top-down pressure from the introduced predator. In small microecosystem patches without the non-native predator, Acari abundance decreased with heat-shock frequency, while Collembola abundance increased. These trends disappeared in larger microecosystem patches or when predators were introduced, although Acari abundance was lower in large patches that underwent heat-shocks and were exposed to the predator. Mean assemblage body length (Collembola) was reduced independently in small microecosystem patches and with greater heat-shock frequency. Our experimental simulation of episodic heatwaves, habitat loss and non-native predation pressure in microecosystems produced evidence of individual and potentially synergistic and antagonistic effects on ecosystem function and microarthropod communities. Such complex outcomes of interactions between multiple stressors need to be considered when assessing anthropogenic risks for biota and ecosystem functioning.

## 1. Introduction

Global biodiversity is undergoing an extinction crisis with declines in the diversity, occurrence and abundance of multiple plant and animal taxa [1–4]. These changes to life on Earth are being driven by multiple anthropogenic pressures (e.g. climate change, habitat loss and degradation, spread of invasive species) [1,4,5] that are, individually or in combination [6–8], profoundly disrupting the biotic communities and ecosystem functions supporting human wellbeing [1,4,9].

Among these pressures, climate change is advancing and becoming one of the pre-eminent direct drivers of anthropogenic changes to the natural world [1,9]. Climate change projections anticipate a rise in land temperature extremes with extreme hot days in mid-latitudes being 3–4°C above current global mean surface temperature and an increased frequency and duration of heatwaves in most terrestrial regions [9]. Such a climate shift is expected to have major impacts on species distribution, abundance and diversity and the ecological interactions maintaining ecosystem function [1,10,11]. Climate change is also likely to result in phenotypic

shifts and altered selection pressure on ecological and physiological traits. For example, as a consequence of metabolic costs versus energetic gains [12,13], extremes of high temperature may elicit fundamental changes to organism body size, e.g. shrinkage versus increase [14–16] or reproductive capacity [17,18]. Although data on the generality of such effects is lacking [19,20], should they occur then the consequences for an organism's survival and functional role would be profound.

Habitat loss is a common feature of land use change or land management intensification that modifies and degrades the functioning of biotic communities and processes at multiple scales [1,21,22]. Climate change is expected to interact with habitat loss to affect biota, potentially by edge effects modifying the temperature within remaining habitat patches [23,24] or by reducing the capacity for compensatory migration and elevating extinction likelihoods by altering population connectivity, microclimate, niche space or trophic interactions [1,7,25,26]. At a global scale, the negative effects of habitat loss have been shown to be exacerbated in geographical areas with the highest maximum temperatures [27], which raises the possibility of synergistic interactions between habitat loss and climate change [9] that further increase rates of biodiversity loss [1,25].

Invasive alien species are another historic and expanding driver of change in biodiversity [1,5,28]. Anthropogenic species introductions into novel habitats have tended to disrupt native biodiversity and ecosystem functions because species within the recipient community lack a coevolutionary history with the invader and hence the necessary adaptations enabling species coexistence [5,28]. Predators exert strong, top-down regulatory pressure on prey populations that can affect communities and functions [29–33]. Among invasive alien species, predators tend to have the greatest impacts, compared to other functional groups, on recipient native communities, particularly in small island ecosystems where prey may experience greater predator encounter rates [5,34]. Moreover, there is evidence that predation pressure can interact with changes in habitat area [35] or environmental temperature [36,37] to modify top-down control of lower trophic levels.

Experiments on model microecosystems are one approach to understand the interplay of multiple global change drivers affecting biodiversity and ecosystem processes [18]. Bryophyte microecosystems are amenable to experimentation because, in addition to the primary producers, they support a community of microarthropods (e.g. Acari, Collembola ≤ 5 mm body length) spanning multiple trophic levels (fungivore, detritivore, predator) that individually operate at very fine spatial scales [38,39]. Such microecosystems allow the controlled manipulation of anthropogenic drivers (e.g. temperature flux, habitat loss, risk of predation) and easier observation of resultant biotic and ecosystem impacts than in larger-scale and more complex ecosystems [31,40–42].

We used a microecosystem experiment to test the individual and combined effects of habitat loss, episodic extreme heat-shock and extreme predation pressure (simulating the introduction of a non-native apex predator) on an ecosystem function (bryophyte chlorophyll fluorescence) and microarthropod communities (abundance, density and body size). We tested three hypotheses (H1–H3) that explore the potential interactive effects of these episodic heat-shocks, introduced predation pressure and habitat loss.

H1: larger microecosystem patches are more resilient to episodic heat extremes because their greater surface area to edge ratio or provision of more microclimatic niches mitigates the effects of environmental heating on photosynthetic function [23,43–45] or microarthropod communities [15,16,40,41]. We, therefore, expected effects of heat-shock episodes to elicit a greater reduction of chlorophyll fluorescence, microarthropod abundance/density and mean body size in the small microecosystem patches. H2: top-down pressure from an introduced apex predator would be reduced in larger ecosystem patches that offer prey species more physical refugia from predator attack [5,35]. We, therefore, expected that greatly elevated predation pressure from the introduction of a voracious generalist predator (*Dalotia coriaria* Kraatz, Staphylinidae) lacking a shared coevolutionary history with the microarthropod prey populations would interact with patch size [35] to reduce prey abundance or densities most severely in smaller ecosystem patches.

H3: the negative effects of extreme heat-shocks, habitat loss and elevated top-down pressure from an introduced apex predator would interact synergistically [24,26,36,37] to compound the reductions in microarthropod abundance, densities and body size.

## 2. Material and methods

### (a) Microecosystem

Ninety-six experimental replicates comprising a bryophyte microecosystem supporting Acari and Collembola communities [40,41] were randomly excised (21 June 2017) using domestic steel circular cookie cutters (110 mm or 50 mm diameter giving microcosms of 95 cm$^2$ and 20 cm$^2$, respectively) from a large, continuous bryophyte sward (*Mnium hornum* Hedw. + rare occurrences of *Polytrichastrum formosum* (Hedw.) G.L. Smith; *Hypnum andoi* A.J.E. Smith) on a brown earth soil in a woodland (Bush Estate, Scotland, UK: Latitude 55.861111, Longitude −3.205833, electronic supplementary material, figures S1 and S2). Each replicate was immediately placed into an individual plastic container (15 cm diameter, 5 cm height) capped by horticultural fleece (electronic supplementary material, figure S3). This fleece enclosed the microcosm and prevented invertebrate migration, but was sufficiently permeable to ensure aeration, the transmission of photosynthetic active radiation for bryophyte photosynthesis and to allow sprays of misted water to penetrate and maintain the moistness of the moss microecosystem within.

### (b) Experimental design

The microecosystem experiment was carried out at the UK Centre for Ecology and Hydrology (UKCEH) Edinburgh (Latitude 55.861111, Longitude −3.205833, electronic supplementary material, figures S1 and S2) over eight weeks (21 June to 11 August 2017). We employed a randomized factorial blocked design with three treatments: (i) microecosystem size (95 cm$^2$, $n = 48$; 20 cm$^2$, $n = 48$); (ii) frequency of heat-shock episodes ('unstressed' controls: $n = 32$; two episodes: $n = 32$; three episodes: $n = 32$); and (iii) addition of an apex predator (present $n = 48$ or control, $n = 48$). We assigned the 96 microecosystems randomly to eight blocks (large plastic trays 100 cm × 50 cm × 16 cm) ensuring each contained a full replicate of the treatment combinations. These were placed outdoors 400 m from the source habitat (electronic supplementary material, figure S2) in a location shaded by trees and buildings. They were situated beneath plastic-covered aluminium mesh workbenches (electronic supplementary material, figure S3) to prevent flooding by rainfall, but otherwise exposed to ambient air movements and temperature (2017: June = 7.9–17.2°C; July = 11.5–19.1°C; August = 11.4–18.9°C).

To keep the moss microecosystem replicates moist they were watered with a fine spray every 24 or 48 h depending on warm weather and immediately after experimental heat-shock episodes (see below) to aid ecosystem recovery.

## (c) Apex predator treatment

To apply a high level of apex predator pressure in the microecosystem simulating that occurring with a non-native species invasion, we introduced a staphylinid beetle known to be an obligate and voracious generalist predator of soil invertebrates [46,47]. *Dalotia* (syn. *Atheta*) *coriaria* Kraatz (Staphylinidae) (electronic supplementary material, figure S4c) is a soil-dwelling predator used for commercial biocontrol because of its efficacy as a generalist predator [46]. It is considerably larger (3–4 mm body length) than most adult and juvenile microarthropods (approx. 0.5–5 mm), and it actively hunts and readily consumes eggs, juvenile or adult stages of many invertebrate orders [46–48]. *Dalotia coriaria* was a good analogue of an invasive non-native predator because, although commercially supplied for glasshouse biocontrol, it is not known to naturally occur in Scotland, aside from a single 2003 record (UK National Biodiversity Atlas: Latitude 57.62525 Longitude –4.11732, Highland region) far from the study location and probably a glasshouse escape. Accordingly, *D. coriaria* lacks a coevolutionary history with the microarthropod fauna of these bryophyte microecosystems, a typical feature explaining the disproportionate impact of invasive non-native predators [5]. Under laboratory conditions, the *D. coriaria* lifespan is 47–60 days for females and males, respectively [49]. Replicates in the predator treatment were inoculated with a single *D. coriaria* (sourced from AGRALAN Ltd, https://www.agralan.co.uk/) at the onset of the experiment and following each heat-shock episode to maintain a consistently high level of introduced predation pressure (i.e. at least one surviving *D. coriaria* in the microecosystem). Given the area of the microecosystems (large = 95 cm$^2$; small = 20 cm$^2$) deployed in this experiment, the application of a single *D. coriaria* beetle per microcosm approximated to a 100- or 500-fold uplift of predation pressure relative to the biocontrol prescription (a single individual per m$^2$ is estimated to consume 10–20 prey items per day—https://www.evergreengrowers.com/atheta-coriaria-8183.html). Coupled to the lack of a coevolutionary history between the apex predator and its prey, this large uplift in potential top-down pressure again mimics the disproportionate impact of a novel alien predator on recipient native communities.

## (d) Episodic heat-shock treatment

Intergovernmental Panel on Climate Change (IPCC) observations and models show climate change is heating terrestrial environments worldwide [9]. In the UK, the latest climate projections anticipate that compared to the 1980–2000 baseline mean summer temperatures in central Scotland for 2060–2079 will be warmer by 0.1–2.8°C for low (IPCC Representative Concentration Pathway (RCP) 2.6) or by 0.6–4.8°C for high (IPCC RCP8.5) emission scenarios (UK Climate Projections, UKCP18, www.metoffice.gov.uk). These projections mean a potential raise in mean summer temperatures in Edinburgh by mid-end of the twenty-first century (1981–2010/RCP2.6/RCP8.5: June = 16.9/19.7/21.7°C; July = 18.8/21.6/23.6°C; August = 18.3/21.1/23.1°C). Extreme weather events, e.g. heat waves are projected to increase in frequency and strength [9]. During the late twentieth and early twenty-first centuries, high temperatures have been recorded in eastern Scotland by the UK meteorological office (www.metoffice.gov.uk) during heatwaves (30.8°C Leuchars, Fife, 2 August 1990; 31.4°C Edinburgh Airport, 4 August 1975; 32.9°C Greycrook, Borders, 9 August 2003).

We challenged the microecosystem replicates with experimental episodic heat-shocks (see below) to simulate short duration heat-waves. Replicates were randomly assigned to acute heat-shock episodes of 2 h duration on two or three occasions during the experiment or to a control (no heat-shock applied). We applied the experimental heat-shock treatment by temporarily and carefully removing the replicate microecosystems from their plastic containers and exposing them for 2 h beneath 40 W light bulbs set within racks of 54 Tullgren extraction funnels (Burkard Scientific Ltd, electronic supplementary material, figure S4a). These extract microarthropods (electronic supplementary material, figure S4b) from soil or litter by producing a temperature gradient and exploiting the behavioural response of microarthropods to descend into the soil or vegetation away from the heat source, where ultimately (if heating is protracted) they fall through the funnel to be collected and preserved in an alcohol containing vessel (70%). Between 2 h heat-shock episodes, we watered the replicates (see above) and left a period of 13 days to allow recovery of the moss microecosystem.

We performed a verification experiment to check that the microecosystem temperature attained under the experimental heat-shock treatments was field realistic given IPCC projections [9] and recorded heatwaves (see above). A 2 h heat-shock raised the mean (±s.e.) surface temperature of the microecosystem ($n$ = 12 of each size class) from the ambient state (i.e. immediately prior to heating treatment: large = 17.9°C ± 0.08; small = 18.1 ± 0.12) to a level of temperature (end of 2 h heat-shock episode: large = 26.5°C ± 0.16; small = 29.5°C ± 0.91). This is greater than RCP8.5 projections of mean summer temperature, but below episodic heatwave temperatures recorded during recent decades (see above examples). Accordingly, we regarded the experimental heat-shock treatments as plausible and commensurate with projected climate change conditions in Scotland under the high emission RCP8.5 scenario [9].

## (e) Microarthropod abundance and body size

All replicates were subjected to a final protracted heating of 24 h duration, an intensity of heat stress that destroyed the moss microecosystem and produced a total extraction of the remaining invertebrate fauna from every treatment. Following the 24 h destructive harvest, we sorted and counted all invertebrates collected to the level of the taxonomic subclass (Acari, Collembola) to provide an estimate of community size (total abundance and density per cm$^2$) for the microecosystem. These two measures of population size give complementary information. Total abundance provided information on the potential carrying capacity of large and small microecosystem patches and how that interacted with the heating and predator treatments; whereas, community density informed on the effect of heating and predation on population size controlling for the influence of variation in habitat area. In addition to the 24 h destructive harvest, we also measured the body length (frons to end of abdomen) of subsamples of Collembola individuals extracted from each large or small microecosystem following application of each heat-shock ($t_1/t_2/t_3$) treatment (Leica DM12.5 microscope, DFC290 camera and Leica Application Suite v. 3.0). Averaging across these temporal subsamples we estimated the mean body length for the Collembola assemblage for each microecosystem ($n$ = 62, excluding unshocked controls) following a total frequency of two or three extreme heat-shocks.

## (f) Chlorophyll fluorescence as an indicator of ecosystem function

To test the impact of the heat-shocks on microecosystem function over the experiment we quantified moss community chlorophyll fluorescence, as an indicator of photosynthetic capacity (photosystem II (PSII) function). We used a Continuous Excitation Chlorophyll Fluorimeter (HandyPEA, Hansatech

Instruments Ltd, UK) on randomly selected bryophyte leaves from each microecosystem, each leaf was dark adapted for 20 min, prior to measurements at a photosynthetic photon flux density of 1500 μmol m$^{-2}$ s$^{-1}$. The rate of chlorophyll fluorescence (Fv/Fm) is calculated as

$$\frac{Fv}{Fm} = \frac{Fm - Fo}{Fm}, \tag{2.1}$$

where Fv is variable fluorescence, Fm is the maximum and Fo is the minimum rate of chlorophyll fluorescence. Fv/Fm is a normalized ratio that works on the principle that the ratio between variable florescence (Fv) and maximal florescence (Fm) approximates the maximum quantum yield of the photosystem (PSII), ranging between 0.75 and 0.84 in healthy mosses [44,50,51], with lower values indicating stress [52]. Chlorophyll fluorescence was measured in all hydrated replicates immediately prior to and following heat-shock episodes. Controls corresponded to the unstressed baseline chlorophyll fluorescence ratio (Fv/Fm) measured in all replicates per block in the unshocked state and was assumed to be unchanged over the short duration of the 2 h shock episode (based on a preliminary trial see the electronic supplementary material, figure S5).

### (g) Statistical analysis

Data were modelled using linear mixed models (LMM) and generalized linear mixed models (GLMM) implemented in R (lme4 function lmer/glmer). The response of microecosystem function (chlorophyll fluorescence) was modelled as the natural log (ln) transformed difference in chlorophyll fluorescence (ln Fv$_{t1}$ − ln Fv$_{t0}$) measured immediately before ($t_0$) and following ($t_1$) each shock treatment (heat-shock controls = mean of all replicates in unshocked state immediately prior to onset of treatment). Modelling differential chlorophyll fluorescence (pre- and post-heat-shock) indicated if the microecosystem was able to maintain its photosynthetic function (no change in ln Fv$_{t1}$ − ln Fv$_{t0}$) or was degraded (negative value) or stimulated (positive value) by the heat-shock episodes. We tested with a LMM the sequential effect of heating episodes on differential chlorophyll fluorescence (ln Fv$_{t1}$ − ln Fv$_{t0}$). Fixed effects fitted were: (i) microecosystem patch size (large or small), (ii) apex predator (+ or −), (iii) the heat-shock episode (1, 2 or 3), and (iv) their two-way interactions.

We verified that the handling of the microcosms did not greatly affect the size of the ecosystem by establishing at the conclusion of the experiment a strong positive relationship between the excised microecosystem patch size (area) and the total mass (LMM estimate = 0.69, $t_{96}$ = 19.90, $p < 0.001$), soil mass (LMM estimate = 0.67, $t_{96}$ = 18.98, $p < 0.001$) and moss biomass (LMM estimate = 0.78, $t_{96}$ = 17.43, $p < 0.001$) measured per microecosystem.

GLMMs (function glmer) of microarthropod abundance following the final heat-shock treatment (24 h duration) were fitted to both Acari and Collembola counts of individuals per microecosystem (Poisson models with log link) and densities per cm$^2$ (Gaussian models with identity link on ln + 1-transformed densities). For Collembola body size, the ln-transformed mean community body length was analysed with a similar GLMM (Gaussian model with identity link). Fixed effects for all models were: (i) microecosystem patch size (large or small), (ii) apex predator (+ or −), (iii) frequency of heat-shock episodes (0, 2 or 3 events of 2 h duration), and (iv) their two-way and three-way interactions.

All models were fitted using maximum likelihood (Laplace approximation) and we report the best fitting model (lowest Akaike information criterion for small sample size (AICc)) from all subsets model comparisons performed using the MuMIn R package. Models fitted experimental block ($n = 8$) as a random effect, while a second random term of 'microecosystem identity'

was fitted to the chlorophyll fluorescence LMM to account for the repeated measures at the replicate level ($n$ observations = 192 over 96 replicates). As a feature of the experimental design, these random effects were retained even when attributable variance was near zero. Statistical significance was assessed with $p$-values ($\alpha = 0.05$) based on asymptotic Wald tests ($P$) for Poisson models and following Satterthwaite's method (function 'lmerModLmer-Test') for Gaussian models (microecosystem chlorophyll fluorescence, microarthropod densities, Collembola body size).

## 3. Results

### (a) Microecosystem photosynthetic capacity

Chlorophyll fluorescence (Fv/Fm) was consistently and negatively affected by episodic heat-shocks to the microecosystem patches. Relative to unshocked control measurements (range 0.75–0.84 indicates normal function), the mean chlorophyll fluorescence ratio was progressively reduced in microecosystems with the sequence of heat-shocks (figure 1a). This was confirmed by the best model of differential chlorophyll fluorescence (ln Fv$_{t1}$ − ln Fv$_{t0}$) over the sequence of heat-shocks (figure 1b; heat-shock 1: $t_{184}$ = −3.237, $p = 0.001$; heat-shock 2: $t_{184}$ = −9.187, $p < 0.001$; heat-shock 3: $t_{184}$ = −12.077, $p \leq 0.001$), indicating a degradation of this microecosystem photosynthetic function.

The rate of differential chlorophyll fluorescence over the sequence of shocks was unaffected by the main effects of microecosystem patch size (estimate: −0.081 ± 0.134, $t_{184}$ = −1.115, $p = 0.266$) and non-native predator presence (estimate: −0.062 ± 0.073, $t_{184}$ = −0.851, $p = 0.396$). Moreover, contrary to our hypothesis (H1), the microecosystem patch size did not modulate the effect of sequential heat-shocks on differential chlorophyll fluorescence as shown by the lack of a statistical interaction (figure 1b; ecosystem size × heat-shock 1: −0.042 ± 0.215, $t_{184}$ = −0.194 $p = 0.846$; ecosystem size × heat-shock 2: −0.252 ± 0.215, $t_{184}$ = −1.172, $p = 0.243$; ecosystem size × heat-shock 3: 0.101 ± 0.248, $t_{184}$ = 0.407, $p = 0.684$).

### (b) Microarthropod abundance, density and body size

All subsets model comparison for GLMMs (AICc) for an abundance of both microarthropod taxa (Collembola and Acari) retained all main effects and their two and three-way interactions (tables 1 and 2). The clearest interpretation of these complex results comes from considering the higher-order three-way interactions between the experimental treatments. Although reduced microecosystem patch size negatively influenced microarthropod abundance, there was a complex interplay with heat-shock and predator treatments (figure 2, tables 1 and 2). In small microecosystem patches without the non-native predator, Acari suffered a progressive decrease in abundance with heat-shock frequency, while Collembola abundance tended to increase (figure 2). These trends were not generally evident in larger microecosystem patches or when predators were introduced, although Acari abundance was reduced in large patches that underwent heat-shocks and were exposed to the non-native predator (figure 2).

Another notable pattern to emerge from this experiment was that two-way interactions between treatments always produced negative effects on microarthropod abundance, yet considering the full interplay between all three treatments produced more heterogeneous outcomes with either positive or negative effects on abundance (tables 1 and 2).

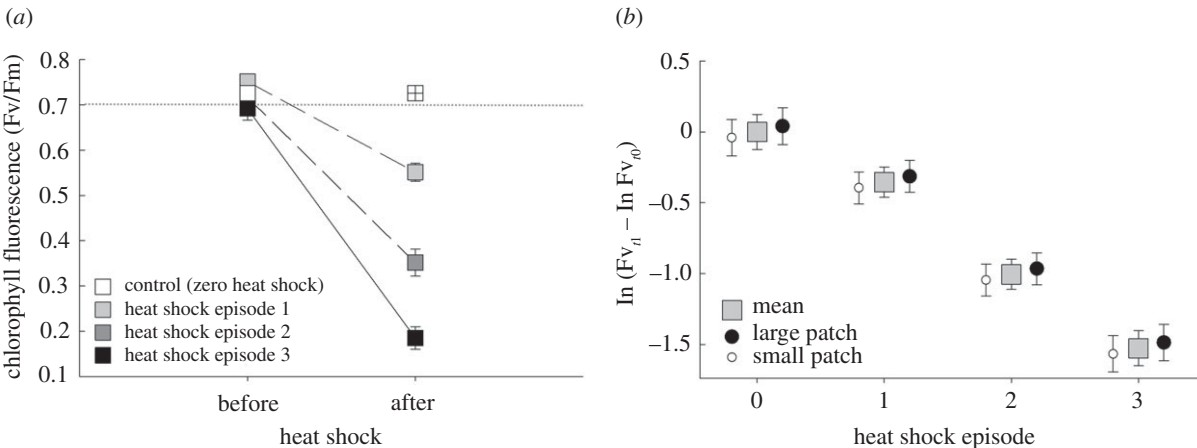

**Figure 1.** The effect of a series of episodic heat-shocks on microecosystem photosynthetic function (chlorophyll fluorescence). (a) The mean ratio of chlorophyll fluorescence (Fv/Fm: the maximum quantum yield of PSII ranges from 0.75 to 0.84 in healthy mosses and indicated by the dotted line) measured before ($t_0$) and following ($t_1$) heat-shock treatments, and (b) the difference in chlorophyll fluorescence rate (In $Fv_{t1}$ — In $Fv_{t0}$) across heating episodes (0/1/2/3) and microecosystem patch sizes (large: 95 $cm^2$, small: 20 $cm^2$). Differential chlorophyll fluorescence indicated unperturbed (no change), degraded (negative value) or stimulated (positive value) photosynthetic function. Data in (a) are raw, untransformed means (±s.e.) of chlorophyll fluorescence ratios, and (b) marginal means (±s.e.) of the difference in chlorophyll fluorescence derived from the final LMM accounting for other model terms. Controls (open square) corresponded to the unstressed baseline ratio (Fv/Fm) derived by measuring chlorophyll fluorescence of all replicates per block in the unshocked state and were assumed to remain stable (crossed open square) over the duration of the 2 h shock episode applied to other replicates (see the electronic supplementary material).

**Table 1.** GLMM for Acari abundance responses to experimental treatments and their interactions. (All subsets model comparison based on AICc was used to determine the best set of fixed effects from the global model (R package MuMIN: function 'dredge', Akaike weight of presented model = 1). $n = 96$ microecosystems arrayed in eight blocks. Experimental block fitted as a random effect. Level of statistical significance $\alpha = 0.05$.)

| predictor of Acari abundance | estimate ± s.e. | z | p |
| --- | --- | --- | --- |
| intercept | 5.64 ± 0.135 | 41.76 | <0.001 |
| apex predator (+) | 0.539 ± 0.026 | 21.05 | <0.001 |
| microecosystem size(small) | −1.620 ± 0.050 | −32.37 | <0.001 |
| heat-shock frequency (0/2/3) | | | |
| two episodes | 0.044 ± 0.028 | 1.537 | 0.124 |
| three episodes | −0.064 ± 0.029 | −2.192 | 0.028 |
| apex predator × heat-shock frequency (two episodes) | −0.616 ± 0.038 | −16.01 | <0.001 |
| apex predator × heat-shock frequency (three episodes) | −0.563 ± 0.039 | −14.32 | <0.001 |
| apex predator × microecosystem size (small) | −0.379 ± 0.067 | −5.628 | 0.001 |
| microecosystem size (small) × heat-shock frequency (two episodes) | −0.095 ± 0.071 | −1.333 | 0.182 |
| microecosystem size (small) × heat-shock frequency (three episodes) | −0.077 ± 0.073 | −1.055 | 0.291 |
| appex predator × microecosystem size (small) × heat-shock frequency (two episodes) | 0.610 ± 0.097 | 6.283 | 0.001 |
| apex predator × microecosystem size (small) × heat-shock frequency (three episodes) | 0.685 ± 0.098 | 6.973 | 0.001 |

Although the response of microarthropod density to the experimental treatments showed similar trends (electronic supplementary material, figure S6), the best subset of models for density retained only a negative effect of microecosystem patch size on Collembola density (estimate = −0.244, $t_{88} = −3.584$, $p < 0.001$). This disparity with the abundance models probably resulted from density removing the strong effect of microecosystem patch size on abundance (tables 1 and 2, figure 2) and as a consequence obscuring other trends in the data.

The mean Collembola body size of the individuals measured in the temporal subsamples obtained following heat-shock applications ($t1/t2/t3$) was 0.95 mm ± 0.62 s.d. ranging from 0.21 to 2.3 mm. The GLMM best subsets model

comparison for log-transformed Collembola assemblage mean body size retained the negative main effects of higher shock frequency (estimate = −0.172, $t_{54.3} = −2.966$, $p < 0.01$) and small microecosystem patch size (estimate = −0.253, $t_{54.4} = −4.418$, $p < 0.001$) (figure 3), but eliminated on the basis of AICc effects of the introduced non-native predator and interactions between treatments. The drop in mean body size occurred following the first heat-shock episode (mean body size ± s.e. = 1.19 ± 0.09 mm) with no change between heat-shock 2 (0.79 ± 0.06 mm) and heat-shock 3 (0.81 ± 0.10 mm). Although the number of Collembola individuals available to be measured per time point varied between replicates (large: 3–88 individuals, mean ± s.d. = 20 ± 19; small: 2–32 individuals, mean ± s.d. = 11 ± 8), including this in the analysis by

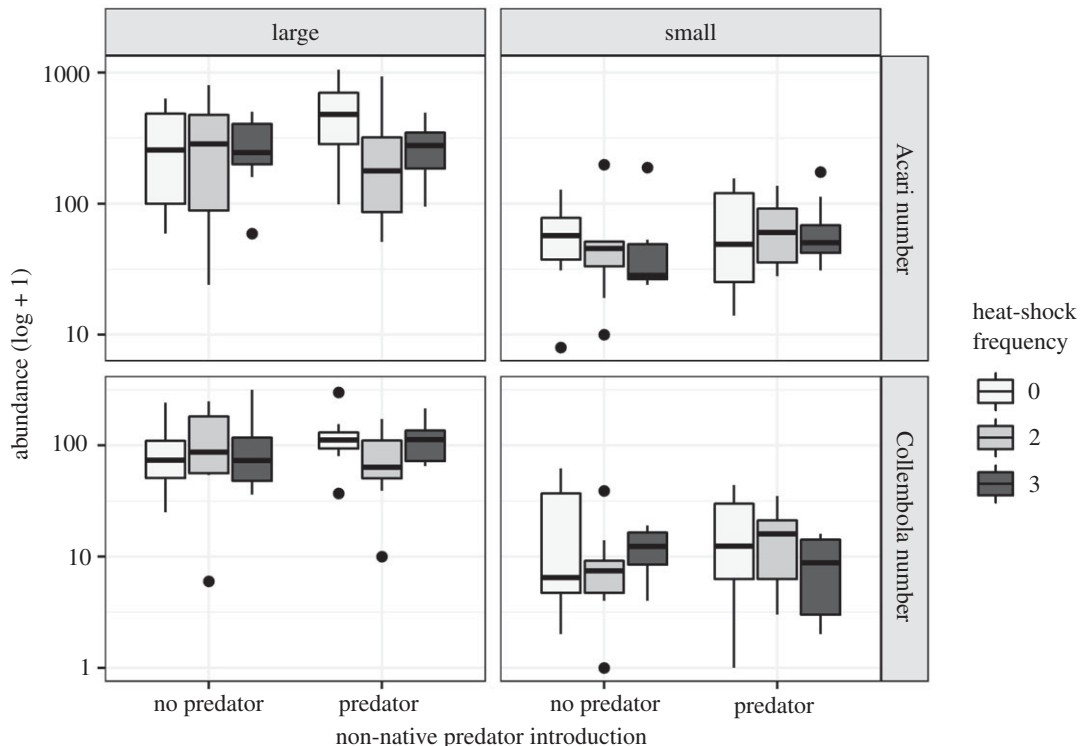

**Figure 2.** Microarthropod (Acari, Collembola) abundance for different experimental levels of microecosystem patch size (large: 95 cm², small: 20 cm²), non-native apex predator presence and frequency of 2 h heat-shock episodes. Boxplots show the medians, interquartile ranges (IQRs) and whiskers (values up to 1.5 IQRs from the box). For effect significances see tables 1 and 2.

**Table 2.** GLMM for Collembola abundance responses to experimental treatments and their interactions. (All subsets model comparison based on AICc was used to determine the best set of fixed effects from the global model (R package MuMIN: function 'dredge', Akaike weight of presented model = 1). $n = 96$ microecosystems arrayed in eight blocks. Experimental block fitted as a random effect. Level of statistical significance $\alpha = 0.05$.)

| predictor of Collembola abundance | estimate ± s.e. | z | p |
|---|---|---|---|
| intercept | 4.554 ± 0.109 | 41.94 | <0.001 |
| apex predator (+) | 0.244 ± 0.047 | 5.141 | <0.001 |
| microecosystem size(small) | −1.621 ± 0.087 | −18.56 | <0.001 |
| heat-shock frequency (0/2/3) | | | |
| two episodes | 0.126 ± 0.049 | 2.598 | 0.009 |
| three episodes | 0.061 ± 0.049 | 1.236 | 0.217 |
| apex predator × heat-shock frequency (two episodes) | −0.592 ± 0.070 | −8.434 | <0.001 |
| apex predator × heat-shock frequency (three episodes) | −0.155 ± 0.067 | −2.310 | 0.021 |
| apex predator × microecosystem size (small) | −0.410 ± 0.127 | −3.225 | 0.001 |
| microecosystem size (small) × heat-shock frequency (two episodes) | −0.862 ± 0.147 | −5.627 | <0.001 |
| microecosystem size (small) × heat-shock frequency (three episodes) | −0.617 ± 0.141 | −4.376 | <0.001 |
| apex predator × microecosystem size (small) × heat-shock frequency (two episodes) | 1.155 ± 0.201 | 5.757 | <0.001 |
| apex predator × microecosystem size (small) × heat-shock frequency (three episodes) | −0.020 ± 0.212 | −0.093 | 0.926 |

weighting the GLMM by number of individuals or including it as a fixed effect had no impact on the final model selected.

## 4. Discussion

The photosynthetic function (bryophyte chlorophyll fluorescence as an indicator of quantum yield of PSII) of these microecosystems was progressively reduced over the series of episodic heat-shocks, indicating a degradation of primary production capacity [50,51]. This finding is consistent with ecophysiological studies that have shown how high temperatures and desiccation affect chlorophyll $\alpha$ fluorescence to disrupt PSII and respiration [43–45]. By repeatedly heating and desiccating the bryophyte tissues, the heat-shock treatments caused photosynthetic disruption, probably owing to a reduction in cell water content. Bryophytes possess the capacity to recover from repeated drying cycles and therefore

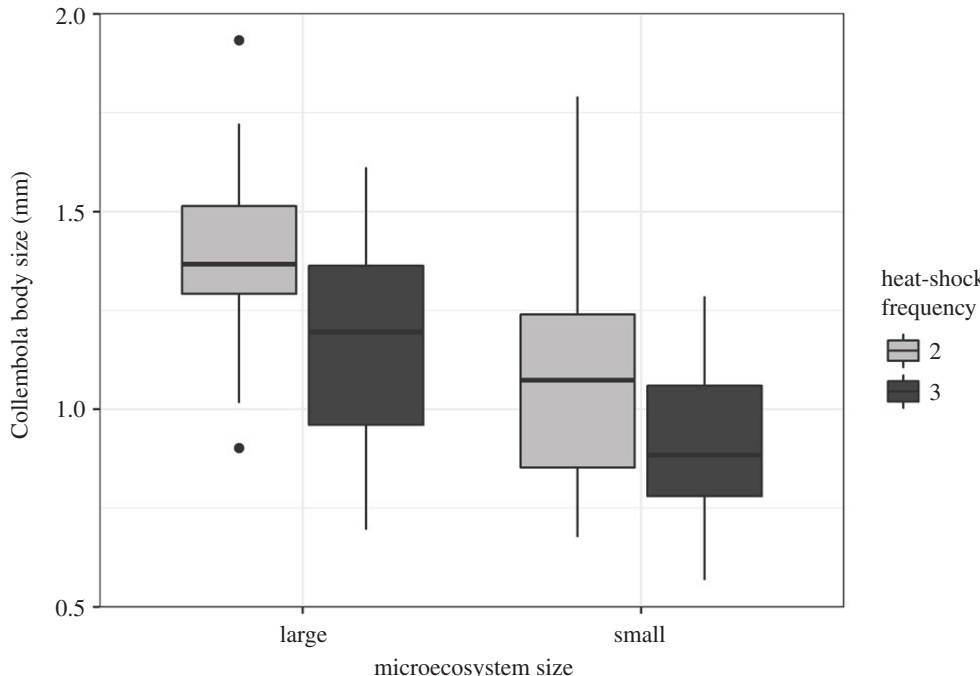

**Figure 3.** Collembola assemblage mean body size for different experimental levels of microecosystem patch size (large: 95 cm$^2$, small: 20 cm$^2$) and frequency of 2 h heat-shock episodes. Boxplots show the medians, interquartile ranges (IQRs) and whiskers (values up to 1.5 IQRs from the box). The GLMM analysis revealed only two statistically significant main effects ($p < 0.01$, see main text).

can tolerate drought [51,53]. However, as a degree of cell degradation occurs in the immediate post-stress recovery period the severity, number and periodicity of stress events determine the degree of this cumulative effect [44]. Contrary to our first hypothesis (H1), however, we found that this direct reduction of microecosystem chlorophyll fluorescence by heat-shocks was not modulated by the size of the microecosystem patch (i.e. no interaction). Furthermore, there was no direct impact of the loss of habitat area (i.e. as a main effect) on photosynthetic function over this experimental timescale. This is in contrast to a habitat fragmentation study (c.f. habitat loss here) that reported effects on biogeochemical functions, e.g. C and N fluxes [31]. Here, the bryophyte community was dominated by a single species and it is possible that a more diverse community will have responded differently to the heat-shock episodes.

Although not a hypothesis a priori in our study, we found no indication that the non-native apex predator treatment affected photosynthetic function. Such an impact on photosynthesis could be postulated to occur via complex trophic interactions in the microarthropod food web [35,54], for example, if predation impacted on herbivores that have the potential to directly modulate plant photosynthesis by consuming leaf tissues or inducing shifts in resource allocation to defence [29,30]. Lacking data on the functional composition of the community meant that we could not test such a hypothesis. Accordingly, we are unable to determine whether the lack of a predator treatment effect on chlorophyll fluorescence was owing to the trophic effects being subtle, unmeasured or simply absent.

As hypothesized (H2 and H3), the interaction between episodic heat-shocks, predator presence and microecosystem size affected microarthropod abundance in complex and sometimes taxon-specific ways. In small microecosystem patches without the non-native predator, Acari decreased and Collembola increased in number in response to heat-

shock frequency; whereas in large patches that underwent heat-shocks and were exposed to the non-native predator, Acari showed a tendency towards reduced abundance, while Collembola were unaffected. Future experiments should examine whether trophic interactions among microarthropod predators (e.g. Acari: Mesostigmata), detritivores and fungivores (e.g. Acari: Oribatidae, Collembola) or food web properties (e.g. modularity) affect the assemblage response to non-native apex predators and their interplay with other stressors [30,35,39,55].

Consistent with island biogeographic and metapopulation theory and evidence [22,40,42], the size of the microecosystem patch, and hence carrying capacity, was important in shaping the interaction with the heat-shock and non-native predator treatments that governed microarthropod abundance. This interaction reflected a meta-analysis showing how climate and habitat loss combine to alter species abundance or diversity [27]. It was also consistent with a previous bryophyte microecosystem experiment that revealed the interaction between environmental temperature and organism dispersal among habitat patches structured microarthropod communities [41]. Differences in ecological traits (e.g. body size, trophic position or functional group) shape an organism's perception of and sensitivity to environmental change [11,56,57]. We can, therefore, hypothesize that the contrasting total abundance responses of Acari and Collembola to the heat-shock × non-native predator × patch size interaction reflected responses by individual microarthropod species occupying different trophic positions (fungivore versus predator) or microhabitat niches [38,39] in the assemblage [41,58]. Time constraints meant we lacked the capacity to obtain the compositional data necessary to disentangle species or trait-based responses to the treatments. Moreover, our microecosystems were completely isolated so we cannot discern the influence of microecosystem connectivity and dispersal processes [21,31,40]. Future studies might examine the effect of multiple

stressor interactions on microecosystems differing in patch fragmentation or isolation, but it should be noted that this is unlikely to modify heat-shock impacts that typically occur at a scale beyond the dispersal ability of many organisms.

Predators often exert substantive top-down pressure on prey populations [29–33], but here compared to the heat-shock and patch size treatments, the presence of the non-native predator had a little overall effect. Predator presence, however, did interact with heat-shock frequency to reduce mite abundance in large patches, while in small patches the predator introduction dampened the impact of the heat-shock × patch size interaction. Predator–prey interactions under environmental heating are likely to be complex with the potential for direct or indirect trophic effects (e.g. cascades or mismatches) that affect ecological dynamics and functioning [33,54]. The cumulative effect of frequent heat-shocks may have had behavioural or physiological (e.g. reproductive capacity) impacts on microarthropods [17,18], which alongside shifts in predator foraging owing to elevated temperature or altered prey availability may have modified the top-down pressure. Further study of the compositional and predator–prey relationships would be needed to elucidate the precise mechanisms, but such an explanation is consistent with other experiments that have shown how predation pressure (mites, centipedes) on microarthropod prey is modified under elevated temperatures [33,36,59,60].

Although the pattern of microarthropod density responses to the three-way interaction of treatments mirrored that of total abundance, statistically there was little effect with only a negative effect of microecosystem patch size on Collembola density. Rather than invalidating the total abundance response, scaling microarthropod abundance ($n$ individuals per cm$^2$) to control for the influence of microecosystem area illustrated the pre-eminence of habitat loss in driving down population size and shaping interactions with other treatments. Indeed while Acari densities scaled linearly, Collembola reductions in density reveal that this taxon was disproportionately impacted by the smaller patch size. The dispersion of microarthropods within the microecosystem may have been altered in response to the treatments (e.g. aggregation in particular microhabitats to avoid predators or high temperatures). If so, this may have been reflected in the total abundance data in ways that were obscured once scaled to densities per unit area, which may have masked the signal of the interactive effects between treatments. Overall it is clear that the influence of microecosystem area on microarthropod abundance was crucial to understand the outcome of multi-stressor interactions for these communities [6–8,25,26].

Collembola assemblage mean body size was reduced by smaller microecosystem patch size and increased heat-shock frequency. This concurs with an earlier study that found an overall drop in mean body size of the collembolan species *Folsomia candida* with temperature treatments [54]. Although unlike Thakur *et al.* [54] and contrary to our hypotheses (H1 and H3), we found no evidence of treatment interactions or an effect of exposure to predation on mean assemblage body size. The observed reduction in mean body size may be a consequence of elevated temperature producing metabolic costs or mismatches for larger consumers [12,13]. Therefore, climate change may drive phenotypic plasticity or selection for smaller consumer body sizes [14,15], although this phenomenon remains to be well established [19,20].

Given the generation time of Collembola is about two to three weeks [61,62] it is possible that selection for smaller body size could have occurred during the experimental timespan (eight weeks). To confirm this would require an energetic and physiological analysis of the sampled individuals or assembling microcosms using laboratory populations of standardized body size [54] to monitor body size evolution over generations. An alternative, but not mutually exclusive, explanation is that in these systems closed to immigration, the heat-shock episodes eliminated the larger, more mobile individuals. This, coupled to the potential production of juveniles within our experimental timeframe [61,62], may explain the observed reduction in assemblage mean body size.

In conclusion, the experimental simulation of three major global change drivers (climate change, habitat loss and introduction of a 'non-native' predator) produced various individual and combined impacts on photosynthetic function (chlorophyll fluorescence) and microarthropod communities in a bryophyte microecosystem. The acute nature of the discrete heat-shock episodes were a major factor impacting both community chlorophyll fluorescence and consumer body size, but modified microarthropod abundance through complex interactions with microecosystem area and non-native predation pressure. This contrasts with the minimal or lack of an effect of chronic warming and drought in microarthropod communities [63] and highlights the potential risks from the cumulative effects on ecosystems from short-term pulse stressors, such as heatwaves predicted to increase in frequency and duration in the future [9]. A notable overall pattern emerging from the analysis of abundance was that two-way interactions between the different stressors (loss of microecosystem area/non-native apex predator/heat-shock frequency) were always negative. This implies that exposure to multiple stressors is potentially synergistic, i.e. disproportionately worse than would be predicted from their (main) effects in isolation [55]. However, accounting for the three-way interaction between the different stressors produced more positive than negative effects, suggesting stressors were antagonistically [55] affecting abundance via different mechanisms that had the effect of cancelling out the impact. This highlights the need to account for as much complexity as possible when assessing multi-stressor impacts on biodiversity in order to improve the accuracy of predicted impacts. This is important because the effects on biota of the interplay between multiple anthropogenic stressors is highly likely to be occurring in nature, but simultaneously remains poorly understood and relatively understudied [1,6–8].

**Data accessibility.** This article has additional data available from the Dryad Digital Repository: https://doi.org/10.5061/dryad.kh1893259 [64].

**Authors' contributions.** A.J.V., A.G. and D.C. conceived the ideas and designed the experiment. C.B. performed the experiment and collected the data. A.J.V. and D.C. performed the statistical analysis and paper preparation. All authors contributed critically to the drafts and gave final approval for publication.

**Competing interests.** The authors declare no competing interests.

**Funding.** We acknowledge the support of the post-graduate internship programme of L'Ecole Nationale Supérieure Agronomique de Toulouse (ENSAT), France, the UKCEH National Capability Resilience project and the ESREA project (ANER Bourgogne-Franche-Comté, France).

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
