## [Peer Review File · Proceedings of the Royal Society B: Biological Sciences]

Review History

RSPB-2021-0032.R0 (Original submission)

Review form: Reviewer 1

Recommendation

Accept with minor revision (please list in comments)

Scientific importance: Is the manuscript an original and important contribution to its field?

Good

General interest: Is the paper of sufficient general interest?

Excellent

Quality of the paper: Is the overall quality of the paper suitable?

Excellent

Is the length of the paper justified?

Yes

Should the paper be seen by a specialist statistical reviewer?

No

Do you have any concerns about statistical analyses in this paper? If so, please specify them explicitly in your report.

No

It is a condition of publication that authors make their supporting data, code and materials available - either as supplementary material or hosted in an external repository. Please rate, if applicable, the supporting data on the following criteria.

Is it accessible?

Yes

Is it clear?

Yes

Is it adequate?

Yes

Do you have any ethical concerns with this paper?

No

Comments to the Author

I thought this was a well written paper on the interacting effects of heat stress (climate change), area (habitat fragmentation), and invasive species on the effect of ecosystem function and community composition in a microecosystem. I do have some questions about your community level analysis, I believe that more depth on the trophic interactions will improve clarity for a more general audience. Please see my detailed comments below.

- 1) I think you need to add more details on the trophic interactions in this system. How many trophic levels are represented, what are they, and what role do Acari and Collembola play in each level, were there any other microarthropods in the system (Line 84, Line 112)? Could predation interactions between Collembola and Acari explain the pattern seen on small area / no predator systems (Line 298-300, Fig. 2)?
- 2) Line 113: Is there any rationale for why you used this particular set of areas?
- 3) Line 133: Was the temperature in the 8 blocks elevated due to the "greenhouse effect" from being under the plastic? Also, did your study site experience any "natural" heat waves during the period of the study?
- 4) Line 143: What was the body size range of microarthropods in your system? This sentence is phrased generally, rather than specific to your study.
- 5) Line 154-156: Unclear if predators were re-added to environments that still had a predator, or if predators were removed and then new ones added after the heatshock treatment. If microenvironments ended up with different numbers of predators, I think this should be accounted for in the analysis. Were predators also re-added to the control treatments that did not receive heatshock?
- 6) Line 166-168: Awkward phrasing, match up the low temperature range with the low IPCC, and the warmer temperature range with the high IPCC.
- 7) Line 186-188: Did microarthropods fall out during heatshock treatments and is this sample what you used to assess counts and body size (Line 209-214)? Unclear in the methods.

- 8) Line 307: Figure S5 that is provided does not match this description. The provided figure shows photosynthesis values in a preliminary trial.
- 9) Line 314 & Fig. 3: Why are there no values for body size from the control treatments? This seems like it would be important information given your discussion on Line 420. You would be able to address it if you just happened to be sampling at a generational cut point, or if the heatshock was selecting for smaller individuals.
- 10) Figure 1B: It is difficult to interpret the error bars between all the overlapping symbols. This figure might be easier to track if all the symbols were offset horizontally.
- 11) Figure 2: The colors cannot be differentiated if it is printed in black and white. Would it make sense to have the colors match those used in Figure 1. Additionally, the green and red colors could prove problematic for the color blind.
- 12) Figure 3: Would it make the life of the reader easier if you used the same color scheme as figure 1 and 2.

Review form: Reviewer 2

Recommendation

Accept with minor revision (please list in comments)

Scientific importance: Is the manuscript an original and important contribution to its field?

Excellent

General interest: Is the paper of sufficient general interest?

Good

Quality of the paper: Is the overall quality of the paper suitable?

Excellent

Is the length of the paper justified?

Yes

Should the paper be seen by a specialist statistical reviewer?

No

Do you have any concerns about statistical analyses in this paper? If so, please specify them explicitly in your report.

No

It is a condition of publication that authors make their supporting data, code and materials available - either as supplementary material or hosted in an external repository. Please rate, if applicable, the supporting data on the following criteria.

Is it accessible?

Yes

Is it clear?

Yes

Is it adequate?

Yes

Do you have any ethical concerns with this paper?

No

Comments to the Author

The authors of the manuscript entitled "Habitat loss, predation pressure and episodic heat-shocks interact to impact arthropods and photosynthetic functioning of microecosystems" performed an experimental test to simulate the effects of climate change, habitat patch reduction and the introduction of an allochthonous predator within mosses. They evaluated the effect on the photosynthetic capacity of mosses and on the structure of the microarthropod community, including the abundance and density of Acari and Collembola and the body size of Collembola. By adopting a randomized factorial blocked design, they were able to demonstrate that the three stressors could act antagonistically on the ecosystem structure thus highlighting the need to account for the complex interactions among multiple environmental factors when evaluating their impacts on ecosystem functioning.

The paper is well written and easy to follow. The topic is timely as the authors are trying to simulate the effects of the main global changes currently occurring worldwide using these 'microcosms' as a model ecosystems. The rationale behind the experiment is clearly stated and all the experimental steps are well described and clearly connected with the hypotheses. The statistical analysis is appropriate and effective. Understanding the results is easy thanks to the nice pictures and tables, and they are discussed point by point in the discussion. The authors are extremely honest in presenting the strengths but also the weakness of their experiment when necessary. Overall, I enjoyed a lot reading this manuscript that I think should be accepted for publication on Proceeding of the Royal Society B.

I have only some minor comments for the authors.

First, I was wondering whether the distribution of microarthropods within moss patches is homogenous or patchy. In other words, it would be nice to know whether your experimental patches started the experiment with more or less the same abundances of Acari and Collembola or whether there were differences among them. Even in case there were differences among patches I do not think that it could hamper the results of the experiment as I guess that the variation among the treatments would be more or less the same. However, if you have this information, for the sake of clarity, I would state it at the beginning of the method section.

Second, I was wondering whether there could be an interaction among the heat shocks and the increasing temperature to which experimental patches were exposed due to the summer season. More in detail, I was wondering whether the environmental temperature could affect the recovery after the heat shocks or if it could make the microcosm more sensitive to the heat shock. Could you provide some consideration about that?

Please, find below also some small suggestions for changes to the text.

Line 208-215: I would specify how you measured the body length of Collembola, e.g. from the top of the frons to the end of the abdomen. Also, I would include the average, minimum and the maximum number of Collembola measured for each patch.

Line 418: I would change "an alternative explanation" into "an alternative explanation (but not excluding the previous one)". In fact, the alternative explanation could also occur simultaneously with the previous one, they do not exclude each other.

Tab. 1 and 2: I would use "< 0.001" for all those p-values that are extremely small. Also, in

Decision letter (RSPB-2021-0032.R0)

04-Feb-2021

Dear Dr Vanbergen,

Thank you for the submission of your manuscript "Habitat loss, predation pressure and episodic heat-shocks interact to impact arthropods and photosynthetic functioning of microecosystems". This has now been peer reviewed and the reviews have been assessed by an Associate Editor. The reviewers' comments (not including confidential comments to the Editor) and the comments from the Associate Editor are included at the end of this email for your reference. As you will see, the reviewers and the Associate Editor are all impressed with this very interesting study (and I agree). However the reviewers have raised some concerns with your manuscript and we would like to invite you to revise your manuscript to address them.

Research ethics:

Use of animals and field studies:

It is a condition of publication that you make available the data and research materials supporting the results in the article. Please see our Data Sharing Policies (<https://royalsociety.org/journals/authors/author-guidelines/#data>). Datasets should be

deposited in an appropriate publicly available repository and details of the associated accession number, link or DOI to the datasets must be included in the Data Accessibility section of the article (<https://royalsociety.org/journals/ethics-policies/data-sharing-mining/>). Reference(s) to datasets should also be included in the reference list of the article with DOIs (where available).

If you wish to submit your data to Dryad (<http://datadryad.org/>) and have not already done so you can submit your data via this link [http://datadryad.org/submit?journalID=RSPB&manu=\(Document not available\)](http://datadryad.org/submit?journalID=RSPB&manu=(Document%20not%20available)), which will take you to your unique entry in the Dryad repository.

Please submit a copy of your revised paper within three weeks. If we do not hear from you within this time your manuscript will be rejected. If you are unable to meet this deadline please let us know as soon as possible, as we may be able to grant a short extension.

Finally, I hope you and your co-authors are well in these challenging times.

Best wishes,
Professor Loeske Kruuk
mailto: proceedingsb@royalsociety.org

Associate Editor
Comments to Author:

This MS presents a very interesting experiment examining the three-way interaction between fragmentation, predators and heat shocks. It's been cleverly designed and implemented and the reviewers and I are all impressed with the experiment and its findings. The reviewers have asked for additional details to be presented, with the most important part being a fuller description of the system being manipulated.

Reviewer(s)' Comments to Author:

Referee: 1

Comments to the Author(s)

I thought this was a well written paper on the interacting effects of heat stress (climate change), area (habitat fragmentation), and invasive species on the effect of ecosystem function and community composition in a microecosystem. I do have some questions about your community level analysis, I believe that more depth on the trophic interactions will improve clarity for a more general audience. Please see my detailed comments below.

- 1) I think you need to add more details on the trophic interactions in this system. How many trophic levels are represented, what are they, and what role do Acari and Collembola play in each level, were there any other microarthropods in the system (Line 84, Line 112)? Could predation interactions between Collembola and Acari explain the pattern seen on small area / no predator systems (Line 298-300, Fig. 2)?
- 2) Line 113: Is there any rationale for why you used this particular set of areas?
- 3) Line 133: Was the temperature in the 8 blocks elevated due to the "greenhouse effect" from being under the plastic? Also, did your study site experience any "natural" heat waves during the period of the study?
- 4) Line 143: What was the body size range of microarthropods in your system? This sentence is phrased generally, rather than specific to your study.
- 5) Line 154-156: Unclear if predators were re-added to environments that still had a predator, or if predators were removed and then new ones added after the heatshock treatment. If microenvironments ended up with different numbers of predators, I think this should be accounted for in the analysis. Were predators also re-added to the control treatments that did not receive heatshock?
- 6) Line 166-168: Awkward phrasing, match up the low temperature range with the low IPCC, and the warmer temperature range with the high IPCC.
- 7) Line 186-188: Did microarthropods fall out during heatshock treatments and is this sample what you used to assess counts and body size (Line 209-214)? Unclear in the methods.
- 8) Line 307: Figure S5 that is provided does not match this description. The provided figure shows photosynthesis values in a preliminary trial.
- 9) Line 314 & Fig. 3: Why are there no values for body size from the control treatments? This seems like it would be important information given your discussion on Line 420. You would be able to address if you just happened to be sampling at a generational cut point, or if the heatshock was selecting for smaller individuals.
- 10) Figure 1B: It is difficult to interpret the error bars between all the overlapping symbols. This figure might be easier to track if all the symbols were offset horizontally.
- 11) Figure 2: The colors cannot be differentiated if it is printed in black and white. Would make sense to have the colors match those used in Figure 1. Additionally, the green and red colors could prove problematic for the color blind.
- 12) Figure 3: Would make the life of the reader easier if you used the same color scheme as figure 1 and 2.

Referee: 2

Comments to the Author(s)

The authors of the manuscript entitled "Habitat loss, predation pressure and episodic heat-shocks interact to impact arthropods and photosynthetic functioning of microecosystems" performed an experimental test to simulate the effects of climate change, habitat patch reduction and the introduction of an allochthonous predator within mosses. They evaluated the effect on the photosynthetic capacity of mosses and on the structure of the microarthropod community, including the abundance and density of Acari and Collembola and the body size of Collembola. By adopting a randomized factorial blocked design, they were able to demonstrate that the three stressors could act antagonistically on the ecosystem structure thus highlighting the need to account for the complex interactions among multiple environmental factors when evaluating their impacts on ecosystem functioning.

The paper is well written and easy to follow. The topic is timely as the authors are trying to simulate the effects of the main global changes currently occurring worldwide using these 'microcosms' as a model ecosystems. The rationale behind the experiment is clearly stated and all the experimental steps are well described and clearly connected with the hypotheses. The statistical analysis is appropriate and effective. Understanding the results is easy thanks to the nice pictures and tables, and they are discussed point by point in the discussion. The authors are extremely honest in presenting the strengths but also the weakness of their experiment when necessary. Overall, I enjoyed a lot reading this manuscript that I think should be accepted for publication on Proceeding of the Royal Society B.

I have only some minor comments for the authors.

First, I was wondering whether the distribution of microarthropods within moss patches is homogenous or patchy. In other words, it would be nice to know whether your experimental patches started the experiment with more or less the same abundances of Acari and Collembola or whether there were differences among them. Even in case there were differences among patches I do not think that it could hamper the results of the experiment as I guess that the variation among the treatments would be more or less the same. However, if you have this information, for the sake of clarity, I would state it at the beginning of the method section.

Second, I was wondering whether there could be an interaction among the heat shocks and the increasing temperature to which experimental patches were exposed due to the summer season. More in detail, I was wondering whether the environmental temperature could affect the recovery after the heat shocks or if it could make the microcosm more sensitive to the heat shock. Could you provide some consideration about that?

Please, find below also some small suggestions for changes to the text.

Line 208-215: I would specify how you measured the body length of Collembola, e.g. from the top of the frons to the end of the abdomen. Also, I would include the average, minimum and the maximum number of Collembola measured for each patch.

Line 418: I would change "an alternative explanation" into "an alternative explanation (but not excluding the previous one)". In fact, the alternative explanation could also occur simultaneously with the previous one, they do not exclude each other.

Tab. 1 and 2: I would use " < 0.001 " for all those p-values that are extremely small. Also, in

Author's Response to Decision Letter for (RSPB-2021-0032.R0)

See Appendix A.

Decision letter (RSPB-2021-0032.R1)

16-Mar-2021

Dear Dr Vanbergen

I am pleased to inform you that your manuscript entitled "Habitat loss, predation pressure and episodic heat-shocks interact to impact arthropods and photosynthetic functioning of microecosystems" has been accepted for publication in Proceedings B.

Data Accessibility section

Open Access

You are invited to opt for Open Access, making your freely available to all as soon as it is ready for publication under a CCBY licence. Our article processing charge for Open Access is £1700. Corresponding authors from member institutions (<http://royalsocietypublishing.org/site/librarians/allmembers.xhtml>) receive a 25% discount to these charges. For more information please visit <http://royalsocietypublishing.org/open-access>.

Paper charges

Thank you for your excellent contribution to Proceedings B. On behalf of the Editors, we look forward to your continued contributions to the journal.

Yours sincerely,
Professor Loeske Kruuk
Editor, Proceedings B
<mailto:proceedingsb@royalsociety.org>

Appendix A

Dear Dr Vanbergen,

Thank you for the submission of your manuscript "Habitat loss, predation pressure and episodic heat-shocks interact to impact arthropods and photosynthetic functioning of microecosystems". This has now been peer reviewed and the reviews have been assessed by an Associate Editor. The reviewers' comments (not including confidential comments to the Editor) and the comments from the Associate Editor are included at the end of this email for your reference. As you will see, the reviewers and the Associate Editor are all impressed with this very interesting study (and I agree). However the reviewers have raised some concerns with your manuscript and we would like to invite you to revise your manuscript to address them.

AUTHORS: Thank you for your positive reaction to our initial manuscript. We have carefully considered and revised the manuscript taking into full account the great majority of the reviewers' comments or explaining the reason why we are unable to do so. The most substantive point, it seemed to us, was the suggestion from referee 1 to consider the potential effect of variable abundance of our introduced apex predator (c.f. presence). You will see below in our response to this referee that we have fully considered this comment and carried out a new analysis (with some logical assumptions) to check our original analysis was robust. This new analysis verified that fitting models accounting for variable 'potential' abundance of the apex predator (microarthropod abundance, density & body size) did not change or only differed qualitatively from our original models (fitting apex predator presence as the predictor). Importantly, the higher order interactions that are the most important effects for the main conclusions of the paper were always unaffected by the switch to fitting potential abundance vs predator presence.

In our opinion, we feel that the original analysis is demonstrated to be robust. We prefer to retain the original approach because: a) it is more justifiable given what we know to be a fact about predator presence (c.f. potential inferred abundance); b) the reanalysis involved assumptions and uncertainties that risk introducing spurious statistical effects; and c) apex predator presence' was the planned factor in our original experimental design. However, we leave it to the Editorial team to guide us on whether they think this alternative analyses should be added to the paper as ESM or not.

Reviewer(s)' Comments to Author:

Referee: 1

Comments to the Author(s)

I thought this was a well written paper on the interacting effects of heat stress (climate change), area (habitat fragmentation), and invasive species on the effect of ecosystem function and community composition in a microecosystem. I do have some questions about your community level analysis, I believe that more depth on the trophic interactions will improve clarity for a more general audience. Please see my detailed comments below.

AUTHORS: thank you for your positive comment, we have done our best to either accommodate your remarks in our revision or justify and explain why it was not possible or prudent to do so.

1) I think you need to add more details on the trophic interactions in this system. How many trophic levels are represented, what are they, and what role do Acari and Collembola play in each level, were there any other microarthropods in the system (Line 84, Line 112)? Could predation interactions between Collembola and Acari explain the pattern seen on small area / no predator systems (Line 298-300, Fig. 2)?

AUTHORS: Because we neither have a baseline survey of the community nor compositional data to show what proportion of the microarthropod fauna were predatory or plant or fungal feeding we cannot know exactly which or how many 'indigenous' trophic levels were present in the system that we experimentally fragmented. However, we can confirm Acari and Collembola were the sole microarthropod taxa collected from the system and therefore we can describe the probable trophic nature of the assemblage to make the level of trophic complexity indigenous to the system clearer to the reader. Accordingly, we have specified the identity of trophic levels likely present in our communities and used existing citation (L83-85: "a community of microarthropods (e.g. Acari, Collembola ≤ 5 mm body length) spanning multiple trophic levels (fungivore, detritivore, predator)..."). We have also added a short paragraph in the Discussion (L362-365 and again mentioned briefly at L378) highlighting, as you suggest, the potential for microarthropod trophic interactions to modulate the response to the apex predator and other stressors.

2) Line 113: Is there any rationale for why you used this particular set of areas?

AUTHORS: The ecosystem fragment size classes were chosen to produce a contrast in ecosystem fragment area that was reasonable given the size and limited mobility of the microarthropods. Furthermore, these sizes were amenable to manipulation: both sizes of fragment could be contained in the plastic tubs and could be readily placed under the Tullgren funnels for the heat shock treatment and microarthropod extraction. Therefore, the choice of ecosystem fragment size represented a balance between biological realism and practical experimental constraints.

3) Line 133: Was the temperature in the 8 blocks elevated due to the "greenhouse effect" from being under the plastic? Also, did your study site experience any "natural" heat waves during the period of the study?

AUTHORS: We did not measure the ambient temperature directly under the plastic coverings so we cannot tell the extent that the temperature underneath was significantly elevated. However, it rarely gets very warm during summer in Scotland and the temperature ranges from a nearby weather station for the experimental period (2017) were: June = 7.9-17.2°C; July = 11.5-19.1°C; August = 11.4-18.9°C (cited in the MS L136). The experimental blocks were situated in the lee of a building and beneath the canopy of an adjacent stand of trees, both of which provided shading from any direct sunlight and mimicked the source habitat conditions (added to revised text L133). Moreover, although topped by plastic fresh ambient air was able to circulate beneath the benches (see Fig. S3 and made clear at L135-136), which would have helped mitigate any artificial greenhouse effect. Even if temperatures were slightly elevated beneath the plastic topped benches, all the replicates/blocks were subject to the same ambient conditions in this very small area. The experimental block was explicit as a random effect in our models to account for any spatial variation in ambient conditions between blocks, even though this was often near to zero it was retained in the model as an experimental design feature. Overall, we think that it was a) unlikely a significant greenhouse effect due to the plastic rain covers occurred; and

b) all replicates experienced the same ambient conditions and blocking accounted for the minimal spatial heterogeneity.

We can also confirm there were no 'natural' heatwaves during the experimental period (it was a typical Scottish 'summer' – see above temperature ranges) apart from the experimental heat shocks applied as a treatment. Finally, all replicates were continually moistened and cooled by watering every 24 or 48 hours depending on need and warm weather (we have made this clearer at L137-139) so that the ecosystem could a) be maintained at similar levels to moisture as encountered in the source habitat and b) recover following heat shocks.

4) Line 143: What was the body size range of microarthropods in your system? This sentence is phrased generally, rather than specific to your study.

AUTHORS: We have added further detail to the revision to address this point. The original sentence in the method is left untouched (“...efficacy as a generalist predator [46], it is considerably larger (3-4 mm body length) than most adult and juvenile microarthropods (0.5-5 mm), and it actively hunts and readily consumes eggs, juvenile or adult stages of many invertebrate orders [46-48].”). But, we add a sentence to the results (L315-317) that reads: “The mean Collembola body size of the individuals measured in the temporal subsamples obtained following heat shock application (t/t2/t3) was 0.95 mm ± 0.62 SD ranging from 0.21-2.3 mm.”

5) Line 154-156: Unclear if predators were re-added to environments that still had a predator, or if predators were removed and then new ones added after the heatshock treatment. If microenvironments ended up with different numbers of predators, I think this should be accounted for in the analysis. Were predators also re-added to the control treatments that did not receive heatshock?

AUTHORS: This is indeed an important point so we have done some reanalysis to check the robustness of our findings. Some background and detailed explanation first. Recall our goal was to elevate predation pressure in the 'predator treatment' above the ambient level in these mesocosms of natural habitat to simulate the impact of the introduction of a non-native apex predator on the community. Because these were not experimentally assembled microcosms, it was not possible to add and remove predator individuals to ensure precisely one invasive predator remained in each replicate because they are small, mobile and adept at staying hidden in the vegetation/soil.

Given our overall objective was to ensure the introduced predation level was significantly elevated relative to their controls we had to make two logical assumptions. First, because *D. coriaria* can fly and are more mobile than the microarthropods they would choose to leave the microecosystems that were unenclosed during the application of the heat shock treatment in the lab. This assumption is supported by the appearance of some *D. coriaria* post-heat shock in the Tullgren extractions (heat shock 1 = 5 individuals out of 32 replicates; heat shock 2 = 7 individuals from 32 replicates; heat shock 3 = 3 individuals from 32 replicates). This reveals some individuals that had burrowed to evade the heat source. Furthermore, we have the anecdotal note that other individuals were found against lab windows (but thus impossible to assign to replicates). Given the potential for replicates to lose the introduced predator, we decided a priori to maintain the level of 'invasive' predation pressure immediately following each heating event by the addition of a single *D. coriaria* to each replicate in the 'predator' treatment subjected to heat shock (as mentioned in the MS L155-159). For the unheated control, because this was a closed system, we made our second assumption that the inoculated original predator was retained until the final 24h heat shock (to extract all the microarthropods from every

replicate). Therefore, we did not repeatedly add *D. coriaria* to unshocked replicates (= heat shock controls within the predator treatment).

Therefore, the reviewer is correct to point out that our approach may have led to differential numbers within and between + predator + heat shocked replicates. To check whether this had a bearing on our results we have re-run our models fitting the variable potential abundance of *D. coriaria* in place of the categorical *D. coriaria* presence/absence parameter. For this analysis, we know that predator controls had zero *D. coriaria*. We assumed that heat shock controls in the predator treatment had 1 x *D. coriaria*. Whereas replicates that received 2 or 3 heat shocks had a theoretical maximum of 3 or 4 *D. coriaria* individuals, respectively, due to the addition of an individual following each heating event. Furthermore, we adjusted these potential maximum counts per replicate where we had knowledge of *D. coriaria* emigration (see above the handful of cases) during the heat shocks. This meant that after accounting for these few replicates, the potential maximum number of *D. coriaria* was adjusted (reduced) so that heat shock control = 1 individual; heat shock frequency of 2 = max 2 or 3 individuals; heat shock frequency of 3 = max of 2, 3 or 4 individuals, depending on the number of recovered (=emigrated) *D. coriaria*.

This parameter of adjusted potential *D. coriaria* abundance was fitted in place of the presence-absence variable and the models rerun. Using 'potential abundance' in place of 'presence/absence' made no overall qualitative difference to the final models of Collembola or Acari abundance obtained. Where quantitative differences in the abundance models occurred, they tended to be seen in the main effects or lower order interactions, but the higher order interactions were unchanged. The likely explanation is that the variability in potential *D. coriaria* abundance was insufficient to produce a response that differed from the simpler presence/absence specification – statistically their effect was the same. For Collembola or Acari densities and for Collembola body size there was no quantitative or qualitative differences between the *D. coriaria* abundance model and the apex predator presence-absence model: they were precisely the same.

Consequently, we prefer to keep the original analysis in the MS. Sticking with the presence-absence parameter is both more conservative and more justifiable given what we know to be a fact about predator presence. We believe this is the best approach because we can only know the 'potential maximum' and not the precise number of *D. coriaria* individuals in each microecosystem. Moreover, although perhaps slim given the lack of major difference between the models, our assumptions and associated uncertainties about the variable numbers of *D. coriaria* may have produced spurious effects that led to some of the slight quantitative changes observed with the substitution of *D. coriaria* presence with the *D. coriaria* potential abundance parameter. Finally, 'apex predator presence' was the factor in our original experimental design and so we prefer to stick with the original analysis to avoid creating unnecessary complexity to this reported experiment.

We supply the tabulated results of this alternative analysis and explanatory text as an extra file to complement this response to the referees. This extra file could be subsequently integrated into the ESM should the editor/reviewer deem it necessary.

6) Line 166-168: Awkward phrasing, match up the low temperature range with the low IPCC, and the warmer temperature range with the high IPCC.

AUTHORS: We corrected this and it reads more smoothly now (L169-172 in the revised MS).

7) Line 186-188: Did microarthropods fall out during heatshock treatments and is this sample what you used to assess counts and body size (Line 209-214)? Unclear in the methods.

AUTHORS: We assessed the overall count of individuals and densities using the final destructive (24h) heat shock of the system at the end of the experiment (note standard use of Tullgren funnels for invertebrate extractions suggest 24-48h for a total sampling of the fauna, as was done here with 24h being sufficient given the small soil fraction). The body size measurements were taken from the subsample of specimens that fell out of the system following each 2-hour heat shock episode. We specify this at L212-218 and hope it is now clearer.

8) Line 307: Figure S5 that is provided does not match this description. The provided figure shows photosynthesis values in a preliminary trial.

AUTHORS: We have checked and amended so the labelling and order of appearance in the text of these suppl. Figures is now correct Fig. S5 = preliminary trial of Chlorophyll PS2 function and Fig. S6 = density responses for Acari and Collembola.

9) Line 314 & Fig. 3: Why are there no values for body size from the control treatments? This seems like it would be important information given your discussion on Line 420. You would be able to address if you just happened to be sampling at a generational cut point, or if the heatshock was selecting for smaller individuals.

AUTHORS: We took advantage of the fact that the 2 h heat-shock episodes produced a subsampling of the microarthropod fauna, which allowed us to obtain data on body size as the experiment proceeded. Because we proceeded in this way there were no body size measurements taken for the heat shock control over the time course of the experiment because, by definition, it required the application of a heat shock to obtain the assemblage subsample for size measurements, which would have invalidated the control. Because of the time constraints of the student performing the work, we were unable to obtain body size measurements on the individuals sampled by the final destructive sampling of the microecosystems, which would have included the heat shock controls. We have added a sentence and some data (L321-323) to show that the shift in the size distribution occurred after the first heat shock (t=1), which suggests that many larger bodied invertebrates were lost to the system at that point.

10) Figure 1B: It is difficult to interpret the error bars between all the overlapping symbols. This figure might be easier to track if all the symbols were offset horizontally.

AUTHORS: We have redrawn this Fig.1B to improve the clarity by adding a standard jitter to the data points on the categorical X-axis scale so that the separate points and error bars can be more readily distinguished from each other

11) Figure 2: The colors cannot be differentiated if it is printed in black and white. Would make sense to have the colors match those used in Figure 1. Additionally, the green and red colors could prove problematic for the color blind.

AUTHORS: We have redrawn all the Figures in the main text and the supplemental files so that they are all white-gray scale and the colour schemes are more consistent between the plots e.g. heat shock episodes are coloured in the same sequence in all plots.

12) Figure 3: Would make the life of the reader easier if you used the same color scheme as figure 1 and 2.

AUTHORS: see above, colour schemes are now matched.

Referee:2

Comments to the Author(s)

The authors of the manuscript entitled "Habitat loss, predation pressure and episodic heat-shocks interact to impact arthropods and photosynthetic functioning of microecosystems" performed an experimental test to simulate the effects of climate change, habitat patch reduction and the introduction of an allochthonous predator within mosses. They evaluated the effect on the photosynthetic capacity of mosses and on the structure of the microarthropod community, including the abundance and density of Acari and Collembola and the body size of Collembola. By adopting a randomized factorial blocked design, they were able to demonstrate that the three stressors could act antagonistically on the ecosystem structure thus highlighting the need to account for the complex interactions among multiple environmental factors when evaluating their impacts on ecosystem functioning.

The paper is well written and easy to follow. The topic is timely as the authors are trying to simulate the effects of the main global changes currently occurring worldwide using these 'microcosms' as a model ecosystems. The rationale behind the experiment is clearly stated and all the experimental steps are well described and clearly connected with the hypotheses. The statistical analysis is appropriate and effective. Understanding the results is easy thanks to the nice pictures and tables, and they are discussed point by point in the discussion. The authors are extremely honest in presenting the strengths but also the weakness of their experiment when necessary. Overall, I enjoyed a lot reading this manuscript that I think should be accepted for publication on Proceeding of the Royal Society B.

AUTHORS: Thank you for your positive assessment of our manuscript.

I have only some minor comments for the authors.

First, I was wondering whether the distribution of microarthropods within moss patches is homogenous or patchy. In other words, it would be nice to know whether your experimental patches started the experiment with more or less the same abundances of Acari and Collembola or whether there were differences among them. Even in case there were differences among patches I do not think that it could hamper the results of the experiment as I guess that the variation among the treatments would be more or less the same. However, if you have this information, for the sake of clarity, I would state it at the beginning of the method section.

AUTHORS: Unfortunately, we do not have this information. As is the case for many organisms, we might assume a patchy distribution for these microarthropods. Because these were not experimentally assembled microcosms but rather a mesocosm experiment using the translocated community fragments, it was impossible to know beforehand the exact abundance, composition or dispersion of the fauna in each mesocosm. Furthermore, a survey of the community in the source habitat to establish a baseline was unfortunately beyond the time and resources of this experiment. Your assumption matches our own, namely, that by excising randomly the moss fragments from the source habitat and then placing these experimental fragments into a fully randomised and blocked factorial experiment meant that we can reasonably expect that there would not be any systematic bias that might confound our treatments.

Second, I was wondering whether there could be an interaction among the heat shocks and the increasing temperature to which experimental patches were exposed due to the summer season. More in detail, I was wondering whether the environmental temperature could affect the recovery after the heat shocks or if it could make the microcosm more sensitive to the heat shock. Could you provide some consideration about that?

AUTHORS: The average ambient temperature did not vary greatly over the time course of the experiment (see ranges detailed above and on L136). The current maritime climate of Scotland means that the summers remain cool and wet and summer temperature extremes are rare at present. Data from the nearby weather station showed the difference in mean ambient temperature during the time course (June/July/August) of the experiment was merely 1.8°C between June and July/August and only 0.1°C between July and August. Given the effect of experimental heating mimicking the climate change scenarios represented an 8-11°C uplift above ambient for small or large microecosystems, respectively (see L193-200), we think that the ambient temperatures were not a major factor in this experiment. Moreover, when the microecosystems were resting between heat shocks all were carefully monitored and regularly (daily or every second day depending on weather) watered to keep the system moist and cool and to aid recovery between heating episodes (L137-139), a measure that would have further mitigated any influence of the ambient temperature and aided recovery from heat shock.

Please, find below also some small suggestions for changes to the text.

Line 208-215: I would specify how you measured the body length of Collembola, e.g. from the top of the frons to the end of the abdomen. Also, I would include the average, minimum and the maximum number of Collembola measured for each patch.

AUTHORS: We have added L212-216: "In addition to the 24 h destructive harvest, we also measured the body length (frons to end of abdomen) of sub-samples of Collembola individuals extracted from each large or small microecosystem following application of each heat-shock (t1/t2/t3) treatment" . We have added (L324-327) the mean +/-SD and range of Collembola individuals sampled and measured from large and small ecosystem patches.

Line 418: I would change "an alternative explanation" into "an alternative explanation (but not excluding the previous one)". In fact, the alternative explanation could also occur simultaneously with the previous one, they do not exclude each other.

AUTHORS: We have added this adjustment as suggested, it now reads L429-433: "An alternative, but not mutually exclusive, explanation is that in these systems closed to immigration, the heat-shock episodes eliminated the larger, more mobile individuals. This coupled to the potential production of juveniles within our experimental timeframe [60, 61] may have produced the observed reduction in assemblage mean body size."

Tab. 1 and 2: I would use "< 0.001" for all those p-values that are extremely small. Also, in

AUTHORS: Changed in the Tables and where statistics are cited in the text.